# Transcriptomics, Epigenetics, and Metabolomics of Primary Aldosteronism

**DOI:** 10.3390/cancers13215582

**Published:** 2021-11-08

**Authors:** Ariadni Spyroglou, George P. Piaditis, Gregory Kaltsas, Krystallenia I. Alexandraki

**Affiliations:** 12nd Department of Surgery, Aretaieio Hospital Athens, Medical School, National and Kapodistrian University of Athens, 11528 Athens, Greece; aspyroglou@gmail.com or; 2Clinic for Endocrinology, Diabetology and Clinical Nutrition, University Hospital Zurich, CH-8091 Zurich, Switzerland; 3Department of Endocrinology and Diabetes Center, G. Gennimatas General Hospital, 11527 Athens, Greece; edk-pgna@otenet.gr; 4Endocrine Unit, First Department of Propaedeutic Medicine, Laiko University Hospital, Medical School, National and Kapodistrian University of Athens, 11527 Athens, Greece; gregory.kaltsas@gmail.com

**Keywords:** primary aldosteronism, transcriptomics, epigenetics, metabolomics

## Abstract

**Simple Summary:**

Improvement in the understanding of the development of primary aldosteronism, the most common cause of endocrine hypertension and mainly caused by aldosterone producing adenomas or hyperplasia, has been continuously accomplished over the past several years. Herein, we summarize the major milestones in the field, including utilization of the newest available molecular techniques to not only shed light on the mechanisms involved in disease development but also to assist in the identification of disease subtypes with distinct laboratory and molecular findings, enabling the personalized treatment of the patients.

**Abstract:**

Introduction: Primary aldosteronism (PA) is the most common cause of endocrine hypertension, mainly caused by aldosterone-producing adenomas or hyperplasia; understanding its pathophysiological background is important in order to provide ameliorative treatment strategies. Over the past several years, significant progress has been documented in this field, in particular in the clarification of the genetic and molecular mechanisms responsible for the pathogenesis of aldosterone-producing adenomas (APAs). Methods: Systematic searches of the PubMed and Cochrane databases were performed for all human studies applying transcriptomic, epigenetic or metabolomic analyses to PA subjects. Studies involving serial analysis of gene expression and microarray, epigenetic studies with methylome analyses and micro-RNA expression profiles, and metabolomic studies focused on improving understanding of the regulation of autonomous aldosterone production in PA were all included. Results: In this review we summarize the main findings in this area and analyze the interplay between primary aldosteronism and several signaling pathways with differential regulation of the RNA and protein expression of several factors involved in, among others, steroidogenesis, calcium signaling, and nuclear, membrane and G-coupled protein receptors. Distinct transcriptomic and metabolomic patterns are also presented herein, depending on the mutational status of APAs. In particular, two partially opposite transcriptional and steroidogenic profiles appear to distinguish APAs carrying a *KCNJ5* mutation from all other APAs, which carry different mutations. Conclusions: These findings can substantially contribute to the development of personalized treatment in patients with PA.

## 1. Introduction

Primary aldosteronism (PA) is the most common cause of endocrine hypertension, with a prevalence of approximately 10% in hypertensive subjects [1,2]. In addition to hypertension and occasionally hypokalemia, aldosterone excess significantly increases cardiovascular risk, stressing the need for better understanding of its pathophysiology for the optimization of treatment strategies [3]. There are two main clinical presentations of PA: aldosterone-producing adrenal adenoma (APA), and bilateral adrenal hyperplasia (BAH), whereas the clinical picture can rarely be attributed to an adrenocortical carcinoma [1]. Recently, both somatic and germline mutations have been identified as causative for the development of APAs; these also affect the clinical phenotype of the disease.

The most frequent genetic alteration in APAs, with a female predominance and a prevalence of 40–50% (and even higher in Asian populations), is a Potassium Inwardly Rectifying Channel Subfamily J Member 5 (*KCNJ5*) mutation which causes depolarization of the membrane of zona glomerulosa (ZG) cells, opening the voltage gated Ca^2+^ channels and increasing Ca^2+^ influx [4,5,6,7]. Acting in a similar way, identified mutations in the ATPase Plasma Membrane Ca^2+^ Transporting 3 (*ATP2B3*) and the Calcium Voltage-Gated Channel Subunit Alpha1 D (*CACNA1D*) genes act by increasing the intracellular Ca^2+^ and stimulating Cytochrome P450 Family 11 Subfamily B Member 2 (aldosterone synthase-*CYP11B2*) expression and subsequent aldosterone synthesis [8,9]. Mutations in the ATPase Na^+^/K^+^ Transporting Subunit Alpha 1 (*ATP1A1*) gene induce cellular acidification due to H^+^ leakage, but the exact mechanism resulting in autonomous aldosterone secretion has not been elucidated yet [8,10]. β-catenin 1 (*CTNNB1*) mutations, identified in a small proportion of APAs, cause constitutive activation of β-catenin and are considered to directly promote CYP11B2 synthesis [11]. More recently, co-existence of *CTNNB1* with G Protein Subunit Alpha Q (*GNAQ*)/G Protein Subunit Alpha 11 (*GNA11*) mutations was documented in 59% of APAs [12]. Rarely, Protein Kinase cAMP-Activated Catalytic Subunit Alpha (*PRKACA*), sporadic Calcium Voltage-Gated Channel Subunit Alpha1 H (*CACNA1H*) and Chloride Voltage-Gated Channel 2 (*CLCN2*) mutations have also been identified in sporadic APAs [13,14,15].

In addition to the rather common somatic mutations responsible for sporadic PA cases, four rare familial forms of the disease associated with early-onset hypertension have been identified. In short, familial hyperaldosteronism type I is attributed to a hybrid Cytochrome P450 Family 11 Subfamily B Member 1 (*CYP11B1*)/*CYP11B2* gene inherited as an autosomal dominant characteristic where aldosterone synthesis is adrenocorticotropic hormone (ACTH)- and not angiotensin II-dependent [16]. Familial hyperaldosteronism type II is caused by a *CLCN2* mutation in chloride channels, clinically expressed as early-onset hypertension along with hypokalemia and was initially described in a population of PA individuals under the age of ten [17]. *KCNJ5* germline mutations are the genetic basis of familial hyperaldosteronism type III [4,18], whereas *CACNA1H* mutations result in a gain of function of the Ca^2+^ voltage gained channel, leading to familial hyperaldosteronism type IV [19,20].

However, the already complex genetic landscape of PA provides a trigger for the further understanding of the pathophysiology of this common endocrine form of hypertension caused by adrenal tumours and/or cancer. Gene expression profiling along with epigenetic and metabolomic studies can elucidate the mechanisms and signaling pathways which have a role in the pathogenesis of PA, enabling the identification of subgroups of PA with distinct clinical, histological, and molecular profiles.

## 2. Methodology

We performed a detailed web-based search of the PubMed and Cochrane database with the terms “Metabolomics”(Mesh) OR “Epigenomics”(Mesh) OR “DNA Methylation”(Mesh) OR “MicroRNAs”(Mesh) OR “Gene Expression”(Mesh) OR “Gene Expression Profiling”(Mesh) AND “Hyperaldosteronism”(Mesh), with the term hyperaldosteronism including “Aldosteronism”, “Conn (‘s) Syndrome” and “primary hyperaldosteronism”, on 25 July 2021. The start date for the literature search was 1 January 1990, and the search was limited to articles written in English and to human studies. Reviews and case reports were excluded from the present analysis. The review was registered on the PROSPERO platform (CRD42021271111). The PRISMA flow diagram can be found in Figure 1.

## 3. Results

### 3.1. Clinical and Histological Traits of PA Patients

The first observations of the distinct characteristics of APAs carrying unique mutations can be obtained from their clinical and histological appearance. APAs carrying *KCNJ5* mutations are significantly larger in size, present lower pre-contrast Hounsfield units in abdominal computed tomography (CT) scans, and histologically display predominantly lipid-rich zona fasciculata (ZF)-like cells [21,22,23]. Further observations associate these tumors with young female patients and higher plasma aldosterone levels [7,24]. On the other hand, APA patients with ATPase mutations are frequently middle-aged men, with hypokalemia and low-renin hyperaldosteronism as well as increased aldosterone responsiveness upon ACTH stimulation, without large adrenal tumors upon CT scan but with histologically well-circumscribed tumors with compact eosinophilic cells and peritumoral hyperplasia [8,24,25]. ZF-like cells appear more typical for *KCNJ5* mutation-containing nodules, and ZG-like cells for *ATP1A1*, *ATP2B3* and *CACNA1D* mutations.

In normal human ZG, in situ hybridization shows focal *CYP11B2* expression with positive cell clusters that according to their size can be characterized as foci, megafoci and larger clusters and which, according to the present histopathology consensus for unilateral PA, are called aldosterone-producing micronodules or APM (formally known as aldosterone-producing cell clusters, or APCCs) [26,27,28]. The presence of APMs has been confirmed in several studies investigating the structure of normal adrenal glands. APMs are composed of both ZG-like and ZF-like cells; however, the ZF-like cells of APMs have high CYP11B2 expression and rather low CYP11B1 and CYP17A1 expression [27,28,29]. APMs do not typically present the cellular atypia seen in APAs [30]; however, a positive correlation between the total APM area and patient age has been documented [31]. In parallel, APMs produce increased levels of aldosterone and 18-oxocortisol, both steroids increased in APAs [32]. It has been suggested that with aging, when the physiological aldosterone production from ZG cells declines, APMs accumulate mutations that can lead to the transition to APAs [32].

Interestingly, APMs in normal adrenal glands are more frequent in women, without ethnic distribution but with a clear correlation with ageing, and often carry known APA mutations such as *CACNA1D* and *ATP1A1* [33]. PA patients with negative adrenal CT scans often present an increased number of APMs, predominantly carrying *CACNA1D* mutations; thus, a potential progression from APMs to micro-APAs can be postulated as part of a continuum in these cases [30,34]. However, in large APAs, which usually carry *KCNJ5* mutations, an APM origin does not seem to be a feasible progression mechanism [30]. Another study found transitional structures with a combination of subcapsular APM-like structure and an inner APA-like microstructure without well-defined borders, characterized by the presence of *KCNJ5* and *ATP1A1* mutations [35]. Recently, two further studies documented that APMs in the adrenals of patients with APAs carried mutations predominantly in *CACNA1D*, but also in *KCNJ5, ATP1A1*, *CACNA1H*, *PRKACA* and *CTNNB1,* weakening the hypothesis that *KCNJ5* mutations do not correlate with the APM-APA transition theory [36,37]. The presence of somatic mutations in APMs suggests that co-driver mutations are necessary in order to promote APA formation. In line with this two-hit theory, co-existence of *CTNNB1* with *GNAQ* or *GNA11* mutations was described in APAs, whereas solitary *GNAQ*/*GNA11* mutations were identified in the adjacent hyperplastic zona glomerulosa of the double mutant APAs [12]. Similarly, the occurrence of *KCNJ5* mutations in adrenals from patients with germline APC Regulator of WNT Signaling Pathway (*APC*) mutations has been previously described [38].

Histological examination of APAs without subtype classification reveals increased nodulation and reduced vascularization in the peritumoral tissues surrounding APAs [27]. The ZG adjacent to APAs appears continuous and thickened, with positive expression of CYP11B2 and Disabled 2 (Dab2), both markers of the ZG, and negative staining for CYP11B1, a typical marker of ZF responsible for cortisol synthesis. This finding is not in line with the observed staining in APMs, which was positive for CYP11B2 but negative for Dab2. Furthermore, the number of APMs in peritumoral adrenal tissues did not differ from control adrenals, whereas the number of megafoci was significantly increased in peritumoral adjacent tissues [27]. In another study, adrenal glands from APA patients presented positive CYP11B2 expression in only one dominant nodule, even in APA cases with histologically documented multinodularity. No conclusions about the correlation between a specific mutation and the multinodularity could be obtained in this study; however, the mutations were always located in the CYP11B2 positive nodules, with multiple positive nodules in the same adrenal gland occasionally carrying different mutations [39,40]. Furthermore, APMs and APAs share immunohistochemical overexpression of the endoplasmatic reticulum protein calmegin (CLGN) [41]. In summary, APMs have high CYP11B2 expression, present a ZG- and ZF-like appearance, and harbor APA-related mutations, all common characteristics with APAs. Still, as the mutational status of the principal nodule in APAs is not necessarily identical with the mutations found in secondary nodules, the theory of an APM to APA transition remains to be elucidated.

An increased number of APMs has been documented in a small cohort of adrenalectomized patients with BAH, predominantly carrying *CACNA1D* mutations [42]. In a different approach, in a very recent study adrenalectomized PA patients with partial or absent biochemical cure that displayed lower lateralization indexes histologically, frequently presented one or more APMs which frequently harbored *CACNA1D* mutations, suggesting common mechanisms in APA and BAH pathogenesis [43].

Furthermore, two-thirds of APAs exhibited positive immunohistochemical staining of G Protein Activated Inward Rectifier Potassium Channel 4 (GIRK4) and Dab2, both markers of the ZG, rendering these possible markers for the distinction of APAs from non-functioning adenomas. Additionally, APAs carrying KCNJ5 mutations exhibited lower GIRK4 expression in APA in comparison to the peritumoral ZG, allowing initial screening for the mutation status of these tumors using immunohistochemistry [44].

### 3.2. Transcriptomics

Gene expression profiles of APAs have increasingly been applied to shed light on the pathophysiology of PA. Either by microarray or serial analysis of gene expression (SAGE), the gene expression profile of APAs is routinely compared to that of adjacent adrenal glands or normal adrenal glands. Thus, an interplay between PA and a variety of signaling pathways can be documented. Among other factors, upon PA a differential expression of several molecules was observed, from classical enzymes involved in steroidogenesis, nuclear receptor transcription factors, ion channels, molecules involved in calcium signaling, and G-coupled proteins to molecules responsible for cell energy, mitochondrial function, protein binding, transcription factors, and oncogenes (Table 1 and Figure 2).

#### 3.2.1. Steroidogenic Enzymes

As the rate limiting step for aldosterone synthesis, *CYP11B2* overexpression is present in a number of studies investigating gene expression profiles in APAs [14,45,46,47,48,49,50,51,52,53,54]. Interestingly, several studies have observed heterogeneity in *CYP11B2* expression in APAs, with one subgroup overexpressed and another group with either unchanged or even reduced CYP11B2 expression [47,55,56,57]. Several studies confirmed that *CYP11B2* expression was significantly higher in tumors carrying *ATP1A1, ATP2B3* or *CACNA1D* mutations than in tumors carrying *KCNJ5* mutations [47,48,49,58]. Kitamoto et al. found increased *CYP11B2* expression in *ATP2B3* tumors but not in *ATP1A1* tumors [59]. Another discrepancy in addition to this initial observation was described by Monticone et al., who documented increased *CYP11B2* expression in APAs with *KCNJ5* mutations [46].

In line with *CYP11B2* expression, differential expression of *CYP11B1*, responsible for cortisol synthesis, has been recognized in several studies. As a common observation, two different *CYP11B1* expression profiles were observed, with a subgroup of APAs presenting an overexpression of this steroidogenic enzyme and a second subgroup displaying very low expression [57]. Interestingly, *CYP11B1* expression was inversely correlated with *CYP11B2* expression. Thus, tumors carrying a *KCNJ5* mutation presented overexpression of *CYP11B1* and concomitant rather low *CYP11B2* levels, whereas *ATP1A1*, *ATP2B3* and *CACNA1D* mutant tumors had very low *CYP11B1* expression along with significant *CYP11B2* overexpression [22,37,49,59]. This pattern is suggestive of a particular biological behaviour of *KCNJ5* tumors, which also appear to co-secrete cortisol [60]. Unlike this rather common finding, a large European multicenter study did not document any significant *CYP11B1* expression differences among the different mutations of APAs [24].

Differential expression of Cytochrome P450 Family 21 Subfamily A Member 2 (*CYP21A2*), the enzyme catalyzing the conversion of progesterone to 11-deoxycorticosterone (a precursor of aldosterone synthesis), is also well documented in PA, with APAs displaying a significant overexpression of this enzyme [48,53]. A concomitant increased expression of Hydroxy-Delta-5-Steroid Dehydrogenase, 3 Beta- And Steroid Delta-Isomerase 2 (*HSD3B2*), the enzyme converting pregnenolone to progesterone, has been documented in the majority of APAs [21,48,59,61]. Interestingly, Cytochrome P450 Family 17 Subfamily A Member 1 (*CYP17A1*) expression in APAs has thus far, shown contradictory trends. In one study, *CYP17A1*, responsible for the hydroxylation of pregnenolone and progesterone, was downregulated in the majority of APAs compared to adjacent adrenal tissue. However, this was not the case in tumors carrying *KCNJ5* mutations, which histologically presented more ZF-like characteristics [21]. Another study documented that *CYP17* expression was downregulated in APAs, without providing information about mutation status [51]. Furthermore, in the same study, Aldo-Keto Reductase Family 1 Member C3 (*AKR1C3*, 17β-hydroxysteroid dehydrogenase type 5) expression showed significantly lower transcript levels in APAs [51]. Finally, an upregulation of Cytochrome P450 Family 11 Subfamily A Member 1 (*CYP11A1*), the catalysator of cholesterol to pregnenolone, was documented in all investigated APAs [56]. For the main steps required for adrenocortical steroidogenesis, see also Figure 3.

#### 3.2.2. Nuclear Receptor Transcription Factors

Several nuclear receptors, acting mainly as transcription factors, have been acknowledged as being involved in aldosterone secretion regulation. In line with these findings, several studies have presented an increase of the Nuclear Receptor Subfamily 4 Group A Member 2 (*NR4A2* or *NURR1*) and Nuclear Receptor Subfamily 4 Group A Member 1 (*NR4A1* or *NGF1B*) transcription factors in APAs [11,22,48], particularly KCNJ5 mutant APAs correlated with a pronounced *NURR1* increase [22,62]. Two further transcription factors play a role in both adrenal development and steroidogenesis, namely Nuclear Receptor Subfamily 5 Group A Member 1 (*NR5A1* or steroidogenic factor-1, *SF-1*) and Nuclear Receptor Subfamily 0 Group B Member 1 (*NR0B1* or dosage-sensitive sex reversal, *DAX-1*); both were found to be significantly increased in APAs [48,63]. In two older studies, however, lower *DAX-1* expression was documented in APAs compared to cortisol-producing or non-functioning adrenal adenomas [54,64]. Finally, in a recent study, the nuclear receptor Retinoic Acid Receptor α, (*RARα*) showed significantly lower expression in APAs compared to normal adrenal glands; the nodulation occurring in APAs was attributed to its downregulation, as this molecule is responsible for normal adrenal zonation [55].

#### 3.2.3. Plasma Membrane Receptors

The single plasma membrane receptor identified so far with a role in APAs is the Scavenger Receptor Class B Member 1 (*SCARB1*), also known as CD36 antigen, responsible for the transport of high-density lipoprotein (HDL) into the ZG cells. In one study, the *SCARB1* expression, important for the cholesterol supplies in the adrenocortical cells, was found significantly upregulated in APAs compared to the adjacent adrenal glands [53].

#### 3.2.4. Ion Channels

Little data is available concerning the differential expression of ion channels in APAs. Concomitant to the expression pattern of CYP11B1 in APAs, and depending on their mutation status as described above, the Potassium Two Pore Domain Channel Subfamily K Member 1 (*KCNK1* or *TWIK-1*) potassium channel and the Solute Carrier Family 24 Member 3 (*SLC24A3*) sodium/calcium exchanger show significant negative correlation with CYP11B1 expression in APAs [21]. The Potassium Two Pore Domain Channel Subfamily K Member 5 (*KCNK5* or *TASK2*) channel is also consistently less expressed in APAs compared to normal adrenal cortex [65]. Recently, Anoctamin 4 (*ANO4*), a calcium dependent chloride channel, was found to be significantly downregulated in APAs compared to normal ZG, independent of their respective mutation status [66]. Expression data on L-type and T-type voltage dependent calcium channels has demonstrated high CACNA1H expression in both normal adrenal glands and APAs, while CACNA1A, CACNA1C and CACNA1E expression was significantly upregulated in APAs [20].

#### 3.2.5. Calcium Signaling

As one of the main pathways promoting physiological aldosterone secretion upon angiotensin II or potassium stimulation is calcium signaling, it is not a surprise that several molecules of the calcium signaling pathway are differentially regulated in APAs. Assié et al. documented an increased expression of Calmodulin 2 (*CALM2*), Calreticulin (*CALR*) and ATPase Sarcoplasmic/Endoplasmic Reticulum Ca^2+^ Transporting 3 (*ATP2A3*, or calcium adenosine triphosphatase 3, *SERCA3*) in APAs compared to the adjacent normal adrenal tissue [53]. Interestingly, in line with the already described heterogeneity of *CYP11B2* expression in two subgroups of APAs, one APA subgroup presents overexpression of Calcium/Calmodulin Dependent Protein Kinase I (*CAMK1*) in parallel with *CYP11B2* overexpression and Calcium/Calmodulin Dependent Protein Kinase II Beta (*CAMK2B*) underexpression, while another group presents the opposite profile [57]. Calneuron 1 (*CALN1*), localized in the endoplasmatic reticulum, binds calcium ions and positively correlates with the increased *CYP11B2* expression in APAs in comparison to non-functioning adrenal adenomas [67]. The endoplasmatic reticulum carrier Calmegin (*CLGN*) is also upregulated in APAs compared to non-functioning adenomas, with a clear positive correlation with *CYP11B2* expression [41,68]. Purkinje Cell Protein 4 (*PCP4*), a molecule modulating calcium binding by calmodulin, has been found to be significantly increased in APAs compared to the adjacent adrenal glands [51]. Finally, Vinisin like 1 (*VSNL1*), a neuronal calcium sensor protein functioning in the transduction of calcium signals, presents significantly higher expression in APAs compared to normal adrenals. Furthermore, *VSNL1* expression in APAs harboring KCNJ5 mutations is significantly higher than in wild-type tumors [52]. Glutathione S-Transferase Alpha 1 (*GSTA1*), an enzyme protecting cells from reactive oxygen species which also serving as transmitter of calcium signaling, presents significantly lower expression in APAs compared to non-aldosterone producing adenomas, while *KCNJ5* mutated APAs had significantly lower expression of this gene than did wild-type APAs [45,69].

#### 3.2.6. G-Protein Coupled Receptors (GPCRs)

Several genes encoding G-protein coupled receptors have been identified as differentially expressed in APAs, whereas a clear interrelation between GPCRs and physiological aldosterone secretion is acknowledged for the Melanocortin 2 Receptor (*MC2R*) and the 5-Hydroxytryptamine Receptor (*5-HTR-4*). In several studies, Luteinizing Hormone/Choriogonadotropin Receptor (*LHCGR*), Gonadotropin Releasing Hormone Receptor (*GNRHR*), 5-HTRs 2A and 4, Angiotensin II Receptor Type 1 (*AGTR1* or *AT1R*), Glutamate Metabotropic Receptor 3 (*GRM3*), Endothelin Receptor Type B (*EDNRB*), *MC2R*, and Prostaglandin E Receptor 1 (*PTGER1*), among others, were all found to be significantly upregulated in APAs [50,70,71,72]. In one recent study, *MC2R* expression correlated positively with that of *AGTR1* in APAs harboring *KCNJ5* and *CACNA1D* mutations, whereas *MC2R* expression correlated positively with Melanocortin 2 Receptor Accessory Protein (*MRAP*) only in *ATP1A1*- and *ATP2B3*-mutated APAs [72]. Moreover, *LHCG*- and *GNRH*-receptor upregulation were both correlated with APAs harboring *CTNNB1* mutations [12,49]. On the contrary, Arginine vasopressin receptor 1A (*AVPR1A*) and Prostaglandin F Receptor (*PTGFR*) were significantly downregulated in APAs [70]. Furthermore, in an ex vivo study using primary cultures from APAs, predominant G-Coupled-Protein Estrogen Receptor 1 (GPER1) expression was documented in these tumors [73,74].

#### 3.2.7. Energy

The cytochrome P450 steroidogenic enzymes require electrons to exert their catalytic activity on cholesterol during the various steps involved in the formation of aldosterone precursors. In accordance with this need, the expression of energy-providing enzymes such as Adrenodoxin (*FDX1*), Cytochrome P450 Oxidoreductase (*POR*), Cytochrome B5 (*CYB5*) have been found to be significantly upregulated in APAs compared to the adjacent ZG or normal adrenal glands [51,53,56]. ATPase Family AAA Domain Containing 3C (*ATAD3C*), a mitochondrial membrane bound ATPase, showed the highest increase in APAs in one study [56], whereas Acyl-CoA Synthetase Short Chain Family Member 3 (*ACSS3*), a gene with acetate-CoA ligase activity, was the top gene upregulated in *KCNJ5* mutant APAs in comparison to wild types in another study [62]. In a recent study, transcriptome data analysis identified alterations in transcriptome signatures in pathways related to mitochondrial fatty acid β-oxidation and peroxisome proliferator receptor-α (*PPARα*), with suppression of ferroptosis suppressor genes and overexpression of genes related to glycolysis/glyconeogenesis in APAs. Furthermore, *KCNJ5* mutated APAs that have a higher proliferative index display increased expression of genes involved in glycolysis and lipid metabolism, an observation reminiscent of the well-characterized role of metabolic reprogramming in cancer progression [75].

#### 3.2.8. Protein Binding

Neurofilament Medium (*NEFM*), which encodes a neurofilament subunit, was significantly upregulated in wild-type APAs for *KCNJ5* mutations (ZG-like APAs) compared to APAs carrying *KCNJ5* mutations (ZF-like APAs). Silencing of NEFM leads to a significant increase of aldosterone secretion in human adrenocortical cell cultures (H295R), suggesting a role of *NEFM* in the physiological negative regulation of aldosterone production [62,76].

Nephronectin (*NPNT*) is a secreted matrix protein with a role in calcium ion binding as well as in integrin binding. *NPNT* was found to be highly overexpressed in APAs with a ZG-like structure carrying *ATP1A1*, *ATP2B3* and *CTNNB1* mutations. *NPNT* production is regulated by the canonical Wnt/β-catenin signaling pathway and may upregulate aldosterone production [47,77].

*PROM1* encodes a transmembrane protein with actinin- and cadherin-binding properties, which also binds cholesterol on the plasma membrane. Prominin 1 (*PROM1*) was found significantly upregulated in APAs when compared to normal adrenal glands [56].

A well-acknowledged mechanism for the development of PA is the constitutive activation of the wnt/β-catenin pathway. In accordance with this, Secreted Frizzled Related Protein 2 (*SFRP2*), a WNT inhibitor, was significantly downregulated in APAs compared to normal adrenal glands or non-functioning adrenal adenomas [11].

#### 3.2.9. Cell Growth/Cell Death

In a SAGE study, although APAs are considered benign tumors, several oncogenes were identified as upregulated in comparison to normal adrenal glands, among others Jun-binding protein (*JAB1*), avian myelocytomatosis viral oncogene (*v-MYC*), IGF-binding protein-2 (*IGFBP2*), teratocarcinoma-derived growth factor (*TDGF1*), and nephroblastoma overexpressed gene (*NOV*). Although *v-MYC, IGFBP2* and *NOV* overexpression was not confirmed by in situ hybridization, no clear conclusions can be made on the mechanisms of tumorigenesis in APAs [53]. The Teratocarcinoma-Derived Growth Factor 1 (*TDGF1*) upregulation in APAs has, however, been confirmed in another microarray study [78]. The apoptosis inhibitors BH3 Interacting Domain Death Agonist (*BID*) and Baculoviral IAP Repeat Containing 2 (*BIRC2*) and 3 (*BIRC3*) were also found overexpressed in a subgroup of APAs harboring *CTNNB1* mutations [49]. Interestingly, the Wnt/β-Catenin pathway is also one of the most frequently altered pathways in adrenocortical carcinomas (ACC), mainly harboring alterations in *CTNNB1*, APC Regulator of WNT Signaling Pathway (*APC*), and Zinc and Ring Finger 3 (*ZNRF3*), suggesting that alterations in this pathway are, in part, shared events in both benign and malignant adrenocortical tumors [79,80].

#### 3.2.10. Immune Response

In a very recent study, microarray analysis of APAs compared to adjacent adrenal cortex identified differentially expressed genes in a series of immune-related pathways, including inflammatory response, interferon-γ response, and IL-6, JAK/STAT3 signaling. APAs presented, in general, significant downregulation of immune related genes, with several of these genes belonging to pathways related to cellular response to oxidative stress, suggesting that oxidative stress may elicit an immune response in the adjacent adrenal cortex. On the contrary, adrenocortical tumor cells appeared to possess mechanisms for counteracting metabolic stress through upregulation of antioxidant systems. APAs were documented to display a high proportion of tumor cells, suggesting that their particular transcriptome profile enables them to escape from immune surveillance [75].

#### 3.2.11. DNA Binding/RNA Polymerase

GATA Binding Protein 6 (*GATA6*), a gene with role in cellular differentiation via activation of HSD3B in the remodeled subcapsular adrenocortical zone, has shown pronounced upregulation in APAs compared to normal adrenals [48]. Paired Related Homeobox 1 (*PRRX1*), a gene related to tumorigenesis encoding a transcription co-activator, was found to be significantly overexpressed in APAs in a microarray study [51]. Dachshund Family Transcription Factor 1 (*DACH1*), a modulator of gene expression and mediator of steroidogenic responses with a role in the wnt/β-catenin pathway, is highly expressed in the ZG and has been identified as a ZG marker and a negative regulator of aldosterone secretion. *DACH1* expression was found to be downregulated in APAs in comparison to normal adrenal glands [56,81]. In functional analyses, it has been shown that DACH1 suppresses aldosterone production; thus, its downregulation is in line with APA development. Brain-Expressed X-Linked 1 (BEX1) is another gene with differential regulation in APAs. In particular, both micro-APAs and APAs present higher BEX1 expression with CACNA1D, ATP1A1 or non-KCNJ5 mutations. This gene is involved in ferroptosis, and it is hypothesized that increasing APA size leads to reduction of the need for anti-ferroptotic mechanisms [25,82].

### 3.3. Epigenetics

The complex regulation of autonomous aldosterone secretion not only includes an altered transcriptional regulation, but also involves further mechanisms such as DNA methylation and the effects of microRNAs.

In general, APAs present hypomethylation of several genes, in part already recognized as presenting transcriptional alterations. The gene with the most frequently hypomethylated promotor is *CYP11B2*, aldosterone synthase. In detail, the CpG island in the promotor region of *CYP11B2* has been found to be hypomethylated in APAs, but not in blood samples of the same patients [83]. Similarly, hypomethylation of *CYP11B2* was not observed in the adjacent adrenal tissue [84,85]. Additionally, the hypomethylated region of *CYP11B2* has not been proven to be induced by the *KCNJ5* or *ATP1A1* mutations [86]. *CYP11B2* hypomethylation in APAs with parallel hypercortisolemia was unchanged; however, these tumors also presented *CYP11B1* promoter hypomethylation, especially at two CpG sites near the Ad1/cAMP response element binding site [87]. Furthermore, lower methylation levels of *CYP11B2* are documented in APAs compared to APMs, suggesting a role of demethylation in a possible APM to APA transition [85].

In addition to these hypomethylated genes, APAs present hypomethylation in other differentially expressed genes, as presented above. In particular, the G-coupled-protein receptors *PCP4, HTR4, MC2R, PTGER1* showed hypomethylation in APAs [71,88]. *PCP4*, one of the genes highly expressed in APAs, presented as one of the most hypomethylated genes in APAs [88]. In a study applying integration of transcriptome and methylome analysis in APAs and the adjacent adrenal gland, 34 genes presented upregulation with parallel CpG hypomethylation. These include aldosterone-related genes (*CYP11B2, MC2R* and hemopexin (*HPX*)) as well as genes related to tumorigenesis (*PRRX*, member RAS oncogene family (RAB38), fibroblast activation protein alpha (*FAP*), Glucosaminyl (N-Acetyl) Transferase 2 (I Blood Group) (*GCNT2*)) and to differentiation (Calmodulin-like Protein 3 (*CALML3*)) [84]. Inversely, hypermethylation of *AVPR1* and Protein Kinase C alpha (*PRKCA*) has been observed in APAs in comparison to normal adrenal glands [83]. Thus, not only is *CYP11B2* hypomethylated in APAs, but several molecules related to *CYP11B2* expression present differential methylation levels as well.

Unlike APAs, ACCs present global hypomethylation when compared to normal and benign tissues. In comparison with benign samples, ACCs present differential methylation status of several CpG sites, including those associated with Insulin Like Growth Factor 2 (*IGF2*) and H19 Imprinted Maternally Expressed Transcript (*H19*), Tumor Protein P53 (*TP53*), and *CTNNB1*. Interestingly, hypermethylation in both ACCs and benign samples has been documented for genes involved in apoptosis and transcriptional and cell cycle control, in particular for Cyclin Dependent Kinase Inhibitor 2A (*CDKN2*), ATA Binding Protein 4 (*GATA4*), Histone Deacetylase 10 (*HDAC10*), PYD And CARD Domain Containing (*PYCARD*), and Secretoglobin Family 3A Member 1 (*SCGB3A1*) [89].

Several microRNAs were identified in APAs as modulators of *CYP11B2* expression and are responsible for the differential regulation of other aldosterone production relevant genes as well. Among others, miR-24 was significantly downregulated in APAs in comparison to normal adrenal glands [90], while its levels were found to be significantly lower in APAs with *KCNJ5* mutations than in those without. In parallel, a significant negative correlation of this microRNA with the expression levels of its target gene, Glutamate Receptor interacting protein 1 (*GRIP1*), has been demonstrated, possibly posing this gene as a candidate factor for aldosterone autonomy [91]. In another study, the expression of miR-375 was significantly downregulated in APAs, whereas the respective in vitro experiments implied a role in tumor suppression acting through the metadherin (MTDH)/Akt pathway [92]. miR-203 exerts an inhibitory action through its target gene, Wnt Family Member 5A (*WNT5A*), and controls aldosterone secretion. miR-203 demonstrated lower expression in APA samples than in adjacent adrenal glands. Interestingly, plasma levels of its target gene, WNT5A, in adrenal vein sampling were found to be useful in differentiating tumor localization and estimating postoperative cure [93]. In a further study, when compared to patients with APAs, patients with bilateral hyperplasia presented overexpression of circulating miR-30e-5p, miR-30d-5p, and miR-7-5p. However, possibly also due to heterogeneity at the microRNA expression level in the APA group, the diagnostic accuracy of these markers does not allow for their application in clinical practice. These findings suggest that APA and BAH form part of a spectrum leading to PA [94]. Finally, miR-23 and miR-34 were found to decrease expression of TASK2 in APAs, leading to an increase in aldosterone production [65].

### 3.4. Metabolomics

One of the oldest approaches to investigating metabolome differences in APAs began decades ago with the initial observation that a patient with APA had elevated C-18-oxygenated steroids [95]. Later studies confirmed the observation that patients with APAs had elevated 18-hydroxycortisol and 18-oxocortisol, while patients with BAH did not present this laboratory phenotype [96]. As a next step, the quantification of these two steroids in the adrenal veins of patients with PA undergoing adrenal vein sampling (AVS) took place and an elevated 18-oxocortisol/cortisol ratio was found, indicating the dominant site in the AVS and allowing differentiation of patients with APAs from patients with BAH [97]. In an attempt to develop a less invasive testing method, urinary 18-hydroxycortisol levels were used with sufficient diagnostic accuracy to distinguish APAs from BAHs [98]. Several metabolic adaptations have since been described in tumorigenesis, with tumor cells undergoing metabolic reprogramming in order to address increased metabolic demands and enhance progression. Characteristic examples include increased glucolysis in cancer cells (the Warburg effect) and the dysregulation of lipid oxidation with increased β-oxidation and subsequent increased NADPH (also critical for adrenal steroidogenesis) with enhancement of CYP11A1 and CYP11B2 activity, possibly leading to increased aldosterone synthesis [75].

The introduction of liquid chromatography with tandem mass spectrometry (LC-MS/MS) in the quantification of adrenal steroids confirmed the previous data, and additionally widened the spectrum of investigated steroids. In addition to the clear elevation of plasma 18-oxocortisol in APAs, increased levels of plasma cortisol, corticosterone, dehydroepiandrosterone (DHEA) and DHEA-S were documented in patients with BAH [99,100]. The combination of peripheral venous steroid profiles with the imaging data from CT or magnetic resonance imaging (MRI) has improved the diagnostic accuracy of correct subtype classification of PA. Lenders et al. observed that the secretion of 18-hydroxycortisol and 18-oxocortisol are highest in familial hyperaldosteronism type 1 and type 3, followed by APAs, whereas BAH patients had comparable levels of these two steroids to patients with essential hypertension [101]. Interestingly, in situ metabolomics has shown that the intratumoral levels of 18-oxocortisol and 18-hydroxycortisol negatively correlate with the CYP11B1 staining [102].

Recently, the steroid profiles of patients with APAs were correlated with their respective genotypes. It has been well documented that APAs carrying *KCNJ5* mutations present significantly higher levels of 18-oxocortisol in both adrenal vein and peripheral plasma samples than all other APAs; wild-type mutations of the *KCNJ5* gene and *KCNJ5* mutant APAs have higher lateralization ratios. In the same study, patients with APAs harboring ATPase mutations displayed the highest peripheral concentrations of aldosterone, cortisol, 11-deoxycorticosterone and corticosterone, while patients with *CACNA1D* mutated APAs had lower concentrations of aldosterone and corticosterone compared to all other groups [103]. In line with the previous observation, another study confirmed that *KCNJ5* carriers display significantly higher levels of 18-hydroxycortisol and 18-oxocortisol when compared to *CACNA1D* carriers. The levels of these hybrid steroids are negatively correlated with CYP11B2 expression, but not with aldosterone levels, and is positively correlated with CYP11B1 expression [37].

Furthermore, the use of peripheral venous plasma steroid profiling in combination with machine learning has not only enabled correct PA subtype classification, but also the correct prediction of APAs with *KCNJ5* mutations, with diagnostic sensitivities of 69% and 85% and specificities of 94% and 97%, respectively. This advance facilitates decision making in *KCNJ5* patients, who benefit most from surgical intervention [104].

Distinct patterns of urinary metabolites were observed in another study, enabling the grouping and distinguishing of essential hypertensives from PA patients and of APA from BAH patients. The identified metabolites include pyrimidine nucleoside and precursors, purine nucleotides and catabolites, and free amino acids [105].

Arlt et al. investigated the urinary steroid profiles of APA patients and documented, in parallel with tetrahydroaldosterone hypersecretion, an increase in glucocorticoid output which was not correlated with any known mutational status. The fact that glucocorticoid output in PA was comparable to that of patients with subclinical Cushing syndrome is particularly striking, suggesting glucocorticoid co-secretion in PA [106].

Targeted metabolomics of blood samples of patients with endocrine hypertension (PA, Cushing syndrome, pheochromocytoma/paraganglioma) and essential hypertension can distinguish between the two groups and has identified four metabolites as being common discriminators of the two disease groups, namely the long-chain acylcarnitines C18:1, C18:2, ornithine, and spermidine [107].

Murakami et al., performing in situ metabolomics, documented distinct molecular signatures between *KCNJ5*- and *CACNA1D*-mutated APAs involving metabolites of steroidogenesis as well as purine metabolism. Activation of purine metabolism was observed in *KCNJ5* mutant APAs, with a significant increase in adenosine monophosphate (AMP) and diphosphate (ADP), whereas these tumors displayed significantly higher 18-steroid intensities [102].

In another study, in situ metabolomics of APMs and APAs identified two distinct APM subgroups, only one of which shared some common characteristics with APAs. This subgroup presented metabolic traits supporting cell proliferation, with increased hexose phosphate and ribose phosphate, and increased purine and tryptophane metabolism. A correlation of these characteristics with respective known mutations was not possible in this study [108].

Finally, a proteomic and phosphoproteomic profiling of APAs in comparison to adjacent adrenal tissue demonstrated that increased steroidogenesis in APA positively correlates with the upregulation of the respective steroidogenic enzymes (*CYP11B2, CYP21A1, HSD3B2*) and their phosphorylation, without any increase in the mitochondrial enzymes providing the energy for the catalyzation of these reactions. Furthermore, the same study identified two distinct protein expression patterns, one common for *KCNJ5* tumors and their adjacent adrenal tissue and another for wild-type APAs for *KCNJ5* and their controls. This study also documented altered extracellular matrix composition in APAs and identified overexpression of Ras Homolog Family Member C (RHOC), an actin-organizing factor, in APAs along with deregulation of the mechanistic target of the rapamycin (mTOR) signaling pathway in these tumors [109].

## 4. Conclusions

In the present study, we have summarized the main findings of transcriptome, epigenetic and metabolomic studies investigating the pathogenesis of primary aldosteronism. One limitation of the present study is that we focused our search only on human studies. Animal and in vitro data, although always useful in elucidating pathophysiological mechanisms, were not included as the combined interpretation of in vivo and in vitro data with different backgrounds (i.e., immortalized cell lines or mouse models with, in part, deviating steroidogenesis patterns) could serve as confounding factors in this already complex pathomechanism.

Taken together, the application of new techniques has importantly contributed to the elucidation of aberrant mechanisms leading to pathological aldosterone production in PA. One main finding in this direction is the identification of APMs as possible APA precursors, as they share several common biological characteristics. However, an APM to APA transition is still a matter of debate. Furthermore, clearly distinct patterns of transcriptional, epigenetic and metabolomic profiling have now been attributed to APAs in comparison to BAHs, most importantly linking APAs with different causative mutations. In particular, two partially opposite transcriptional and steroidogenic profiles can distinguish APAs carrying a *KCNJ5* mutation from all other APAs, including those carrying a *CACNA1D*, *ATP1A1*, *ATP2B3* or even *CTNNB1* mutation. Interestingly, recent studies have analyzed the distinct metabolic signatures of these different mutations in depth. These findings can substantially contribute to the development of personalized treatments in patients with PA caused by adrenal neoplasms or hyperplasia. Although major progress has been made in understanding APAs, much remains to be done, as the molecular profiles and pathophysiological mechanisms underlying the development of BAH have not been sufficiently clarified yet.

## Figures and Tables

**Figure 1 cancers-13-05582-f001:**
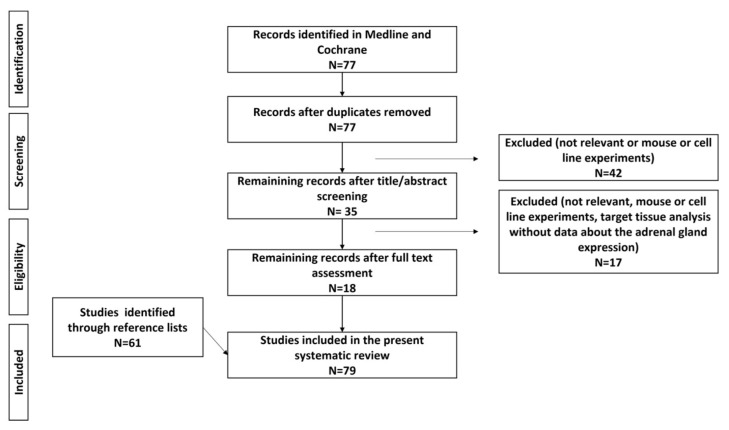
Flowchart for data collection.

**Figure 2 cancers-13-05582-f002:**
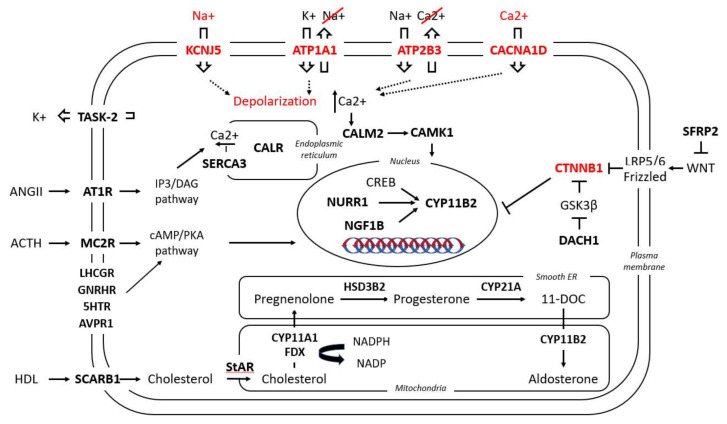
Simplified presentation of the main pathways involved in aldosterone regulation under physiological conditions and in APAs. In bold are the molecules involved in these pathways which have been found to be up-/down-regulated in APAs; in red are the five known mutant genes responsible for APAs and their aberrant cellular function. For nomenclature, the name of the respective genes and proteins have been used; for their respective abbreviations, see Table 1.

**Figure 3 cancers-13-05582-f003:**
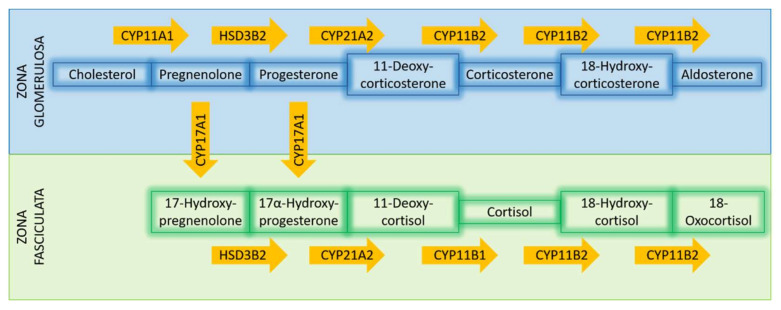
Simplified presentation of steroidogenesis in the zona glomerulosa and zona fasciculata; the yellow arrows represent the enzymes which catalyze the respective reactions.

**Table 1 cancers-13-05582-t001:** Transcriptome-identified genes up- (↑) or downregulated (↓) in the following conditions: (a) in APAs versus normal adrenal tissue or other adrenal adenomas; (b) in two different subgroups of APAs: ZF-like (in some studies defined as carrying KCNJ5 mutations) versus ZG-like (in some studies defined as either WT or as carrying the ATP1A1, ATP2B3, CACNA1D or CTNNB1 mutations).

Genes	Description	Trend
** Steroidogenic enzymes **		
*CYP11B2*	Cytochrome P450 Family 11 Subfamily B Member 2	↑(a), ↔ ↑(b)
*CYP11B1*	Cytochrome P450 Family 11 Subfamily B Member 1	↑(b)
*CYP21A2*	Cytochrome P450 Family 21 Subfamily A Member 2	↑(a)
*HSD3B2*	Hydroxy-Delta-5-Steroid Dehydrogenase, 3 Beta- And Steroid Delta-Isomerase 2	↑(a)
*CYP17A1*	Cytochrome P450 Family 17 Subfamily A Member 1	↓ (a), ↑(b) *
*CYP11A1*	Cytochrome P450 Family 11 Subfamily A Member 1	↑(a)
*AKR1C3*	Aldo-Keto Reductase Family 1 Member C3	↓(a)
** Nuclear receptors/transcription factors **		
*NR4A2*	Nuclear Receptor Subfamily 4 Group A Member 2 (NURR1)	↑(a), ↑(b)
*NR4A1*	Nuclear Receptor Subfamily 4 Group A Member 1 (NGF1B)	↑(a)
*NR0B1*	Nuclear Receptor Subfamily 0 Group B Member 1 (DAX1)	↑(a) *
*NR5A1*	Nuclear Receptor Subfamily 5 Group A Member 1 (Steroidogenic factor 1 _ SF1)	↑(a)
*NR1B1*	Retinoic Acid Receptor Alpha (RARα)	↓(a)
** Plasma membrane receptor **		
*SCARB1*	Scavenger Receptor Class B Member 1 (CD36)	↑(a)
** Ion channels **		
*KCNK1*	Potassium Two Pore Domain Channel Subfamily K Member 1 (TWIK-1)	↓(b)
*KCNK5*	Potassium Two Pore Domain Channel Subfamily K Member 5 (TASK-2)	↓(a)
*SLC24A3*	Solute Carrier Family 24 Member 3 (sodium calcium exchanger)	↓(b)
*ANO4*	Anoctamin 4 (calcium dependent chloride channel)	↓(a)
*CACNA1A*	Calcium Voltage-Gated Channel Subunit Alpha1 A	↑(a)
*CACNA1C*	Calcium Voltage-Gated Channel Subunit Alpha1 C	↑(a)
*CACNA1E*	Calcium Voltage-Gated Channel Subunit Alpha1 E	↑(a)
** Calcium signaling **		
*CALM2*	Calmodulin 2	↑(a)
*CALR*	Calreticulin	↑(a)
*CAMK1*	Calcium/Calmodulin Dependent Protein Kinase I	↑(b)
*CAMK2B*	Calcium/Calmodulin Dependent Protein Kinase II Beta	↓(b)
*CALN1*	Calneuron 1	↑(a)
*ATP2A3*	ATPase Sarcoplasmic/Endoplasmic Reticulum Ca^2+^ Transporting 3 (SERCA3)	↑(a)
*CLGN*	Calmegin	↑(a)
*PCP4*	Purkinje Cell Protein 4	↑(a)
*VSNL1*	Visinin Like 1	↑(a), ↑(b)
*GSTA1*	Glutathione S-Transferase Alpha 1	↓(a), ↓(b)
** G-protein coupled receptors **		
*LHCGR*	Luteinizing Hormone/Choriogonadotropin Receptor	↑(a) **
*GNRHR*	Gonadotropin Releasing Hormone Receptor	↑(a)
*HTR2A*	5-Hydroxytryptamine Receptor 2A	↑(a), ↑(b)
*HTR4*	5-Hydroxytryptamine Receptor 4	↑(a)
*AGTR1*	Angiotensin II Receptor Type 1 (AT1R)	↑(a), ↑(b)
*PTGER1*	Prostaglandin E Receptor 1	↑(a)
*GRM3*	Glutamate Metabotropic Receptor 3	↑(a)
*EDNRB*	Endothelin Receptor Type B	↑(a)
*MC2R*	Melanocortin 2 Receptor	↑(a), ↑(b)
*AVPR1A*	Arginin Vasopressin Receptor 1A	↓(a)
*PTGFR*	Prostaglandin F Receptor	↓(a)
*GPER1*	G-Protein-Coupled Estrogen Receptor 1	↑(a)
** Energy **		
*FDX1*	Adrenodoxin	↑(a)
*POR*	Cytochrome P450 Oxidoreductase	↑(a)
*CYB5*	Cytochrome B5 Type A	↑(a)
*ATAD3C*	ATPase Family AAA Domain Containing 3C	↑(a)
*ACSS3*	Acyl-CoA Synthetase Short Chain Family Member 3	↑(b)
-	Genes related to lipid metabolism, glycolysis, and antioxidant systems	↑(a)
** Protein binding **		
*NEFM*	Neurofilament Medium Chain	↓(b)
*NPNT*	Nephronectin	↓(b)
*MRAP*	Melanocortin 2 Receptor Accessory Protein	↑(a), ↑(b)
*PROM1*	Prominin 1	↑(a)
*SFRP2*	Secreted Frizzled Related Protein 2	↓(a)
** Cell growth/cell death **		
*COPS5*	COP9 Signalosome Subunit 5 (JAB1)	↑(a)
*MYC*	MYC Proto-Oncogene, BHLH Transcription Factor	↑(a)
*IGFBP2*	Insulin Like Growth Factor Binding Protein 2	↑(a)
*CCN3*	Cellular Communication Network Factor 3 (IGFBP9 or NOV)	↑(a)
*TDGF1*	Teratocarcinoma-Derived Growth Factor 1	↑(a)
*BID*	BH3 Interacting Domain Death Agonist	↓(b) **
*BIRC2*	Baculoviral IAP Repeat Containing 2	↓(b) **
*BIRC3*	Baculoviral IAP Repeat Containing 3	↓(b) **
** Immune response **		
-	Genes related to inflammatory response, interferon-γ response, and IL-6, JAK/STAT3 signaling	↓(a)
** DNA binding/RNA polymerase **		
*GATA6*	GATA Binding Protein 6	↑(a)
*PRRX1*	Paired Related Homeobox 1	↑(a)
*DACH1*	Dachshund Family Transcription Factor 1	↓(a)
*BEX1*	Brain-Expressed X-Linked 1	↑ (a), ↓(b)

* Ambiguous results, please see text for details. ** Higher expression in CTNNB1 tumors.

## Data Availability

Not applicable.

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
