# Peer review of "Transcriptomics, Epigenetics, and Metabolomics of Primary Aldosteronism"

_cancers, 2021, doi:10.3390/cancers13215582_

Round 1

Reviewer 1 Report

Spyroglou and authors present a systematic review on omics and epigenetic alterations associated with primary aldosteronism. This is a comprehensive, organized, and informative paper with good figures and tables.

There are a few comments.

Comments

1) What were the start and end dates for manuscripts included in literature search?

2) Take care to italicize gene names but not protein names. Suggest a check throughout.

3) Introduction: ATP1A1 mutations cause intracellular acidification rather than calcium increased calcium concentration (Stindl Endocrinology 2015: PMID: 26418325). This study should be clarified, and the paper cited.

4) Introduction: CTNNB1 mutations: Morris Brown´s group recently reported co-existing GNAQ/11 mutations in 59% of APAs with a CTNNB1 mutation (Zhou J, Nat Genetics 2021: PMID: 34385710). Please also cite.

5) Sporadic CACNA1H mutations in APAs have also been identified and this should be mentioned and cited (Nanba K, Hypertension 2020: PMID: 31983310)

6) Why particularly highlight the association of FH-II link with early onset hypertension? It reads as though the other forms were not. Is this so? I don’t think so: the genetic studies of Ute Scholl identifying FH variants were performed by screening young individuals (< 10 years) with hypertension. Please clarify that familial forms are mostly present with early onset HTN.

7) Section 3.1, 2nd paragraph: the Munich/Sendai groups led an international consensus study for improved histopathology nomenclature in PA (Williams JCEM 2021: PMID: 32717746). Accordingly, please change APCC to “aldosterone-producing micronodules (APMs, formally known as aldosterone-producing cell clusters) in accordance with this guideline that should be cited. And thereafter, only refer to APMs.

8) Steroid profiling and genotype. The use of steroid profiling using LC-MS/MS combined with machine learning has been validated for KCNJ5 mutation prediction (Eisenhofer JAMA Network 2020. PMID: 32990741). This should be mentioned and cited.

9) Page 12, paragraph beginning Murakami et al. another study using the same technique (and I believe the same group) investigated metabolomic phenotypes of aldosterone-producing micronodules (APMs) and found evidence for progression of a specific subset of APMs to APAs (Sun N, Hypertension 2020: PMID: 31957522). This study should be cited and included in the literature search (flow chart for data collection).

Typos:

Section 3.1: CACNAsA1D => Typo?

Bottom page 6: CTNNB => CTNNB1

Page 11, paragraph 3 : “Lenders et al. … 18-hydroxycortisol

Author Response

Reviewer 1:

Spyroglou and authors present a systematic review on omics and epigenetic alterations associated with primary aldosteronism. This is a comprehensive, organized, and informative paper with good figures and tables.

There are a few comments.

We thank the Reviewer for his overall positive comments

Comments

1)What were the start and end dates for manuscripts included in literature search?

Start date was the 1st January 1990, stop date was the 25th July 2021. Only articles written in English and referring to human studies were included in the manuscript (p. 3; lines 97-99).

2) Take care to italicize gene names but not protein names. Suggest a check throughout.

As suggested by the Reviewer, gene names are now italicized throughout the manuscript.

3) Introduction: ATP1A1 mutations cause intracellular acidification rather than calcium increased calcium concentration (Stindl Endocrinology 2015: PMID: 26418325). This study should be clarified, and the paper cited.

We would like to thank the Reviewer for this remark. The respective changes have now been included in the present version of the manuscript (p. 2, line 63-66).

4) Introduction: CTNNB1 mutations: Morris Brown´s group recently reported co-existing GNAQ/11 mutations in 59% of APAs with a CTNNB1 mutation (Zhou J, Nat Genetics 2021: PMID: 34385710). Please also cite.

As suggested by the Reviewer, this new information was included in the manuscript (p. 2, line 68-70).

5) Sporadic CACNA1H mutations in APAs have also been identified and this should be mentioned and cited (Nanba K, Hypertension 2020: PMID: 31983310)

Following the Reviewer’s suggestion, we have now added the information/citation in the text (p. 2, line 71).

6) Why particularly highlight the association of FH-II link with early onset hypertension? It reads as though the other forms were not. Is this so? I don’t think so: the genetic studies of Ute Scholl identifying FH variants were performed by screening young individuals (< 10 years) with hypertension. Please clarify that familial forms are mostly present with early onset HTN.

We fully agree with the Reviewer’s comment, and we have now corrected this statement according to his suggestion. Still, we left the “definition” of FH-II as described by Ute Scholl (p. 2, line 75).

7) Section 3.1, 2nd paragraph: the Munich/Sendai groups led an international consensus study for improved histopathology nomenclature in PA (Williams JCEM 2021: PMID: 32717746). Accordingly, please change APCC to “aldosterone-producing micronodules (APMs, formally known as aldosterone-producing cell clusters) in accordance with this guideline that should be cited. And thereafter, only refer to APMs.

We would like to thank the Reviewer for clarifying the proper use of the terms. The appropriate nomenclature is now applied throughout the manuscript.

8) Steroid profiling and genotype. The use of steroid profiling using LC-MS/MS combined with machine learning has been validated for KCNJ5 mutation prediction (Eisenhofer JAMA Network 2020. PMID: 32990741). This should be mentioned and cited.

As suggested by the Reviewer, this important study is now summarized and cited in the present version of the manuscript (p. 13, lines 523-527).

9) Page 12, paragraph beginning Murakami et al. another study using the same technique (and I believe the same group) investigated metabolomic phenotypes of aldosterone-producing micronodules (APMs) and found evidence for progression of a specific subset of APMs to APAs (Sun N, Hypertension 2020: PMID: 31957522). This study should be cited and included in the literature search (flow chart for data collection).

As recommended by the Reviewer, the findings of this study were added in the manuscript and the respective reference is now included in the flow chart (now, p. 14, lines 548-553 and Figure 1).

Typos:

Section 3.1: CACNAsA1D => Typo?

This was indeed a typo and is now corrected.

Bottom page 6: CTNNB => CTNNB1

Here again, the typo was corrected.

Page 11, paragraph 3 : “Lenders et al. … 18-hydroxycortisol

Here again, the typo was corrected.

Reviewer 2 Report

In this review article Spyroglou and coll. summarized the main findings in the fields of transcriptomics, epigenetics and metabolomics of primary aldosteronism.

The review is well written and quite comprehensive, but it has great limitations that prevent consideration in a high IF journal.

The review provides a summary of findings rather being a critical view of the current state of knowledge. Of course, this is a great limitation.

The review is expected not only to report the presence of APCC, but also mention the ongoing debate on their possible contribution to the development of hyperplasia and/or APA. No effort was done to discuss CYP11B1 expression in the APAs, hypo-or hypermethylation of CYP11B2, modulation of gene expression by microRNAs, PRRX1, DACH1, and other factors. Figures do not add any relevant information. After reading the review, the reader is left with a huge amount of spotted information but no critical general view.

Studies that used cell lines were excluded from the literature search. Although this exclusion criterion was aimed at restricting the search to the findings from human tissues, it leads to the exclusion of studies that are crucial to offer a mechanistic interpretation of what observed in the human adrenal tissues. Hence, their findings should be integrated with those reported in the review.

Age-related differences are ignored. Similarly, receptors as GPER were not considered at all.

Author Response

Reviewer 2:

In this review article Spyroglou and coll. summarized the main findings in the fields of transcriptomics, epigenetics and metabolomics of primary aldosteronism.

The review is well written and quite comprehensive, but it has great limitations that prevent consideration in a high IF journal.

The review provides a summary of findings rather being a critical view of the current state of knowledge. Of course, this is a great limitation.

We appreciate the Reviewer’s assessment. Our initial intention was to present the available studies in this evolving field also considering some contradictory findings. We in particular focused on the patterns of the distinct genetic alterations observed in the development of APAs and their correlation with the respective mutational status due to the rather limited amount of evidence available. In the revised version of the manuscript, we have made an attempt to be more critical while presenting current knowledge and controversies. Main findings are now commented on, to provide possible explanations for the origin and development of PA.

The review is expected not only to report the presence of APCC, but also mention the ongoing debate on their possible contribution to the development of hyperplasia and/or APA.

We would like to thank the Reviewer for his suggestion. In the present version of the manuscript, the APCC definition was amended appropriately to adapt to the histopathology consensus definition. Furthermore, studies investigating the potential of APCC transition in APAs are now discussed (see results p. 3-4 , lines 119-179 and p. 14, lines 548-553).

No effort was done to discuss CYP11B1 expression in the APAs, hypo-or hypermethylation of CYP11B2, modulation of gene expression by microRNAs, PRRX1, DACH1, and other factors.

As indicated by the Reviewer, in the revised version of the manuscript parts, which were insufficiently discussed, are now expanded to provide comprehensive information about the respective topics (p. 5, lines 216-220, p. 7, lines 282-285, p. 8, lines 328-330, p. 9, lines 397-412, p. 11-12, lines 431-478, p. 13, lines 518-527 and p. 14, lines 548-553).

Figures do not add any relevant information.

We appreciate the Reviewer’s comment. When designing the figures, we were aiming at summarizing the main mechanisms identified in aldosterone regulation, to assist the reader to spot the respective change while reading the manuscript. In this context, figures only depict parts of the information described in the manuscript and could be omitted from the main text but provided as supplemental material if the Editor suggests removing them.

After reading the review, the reader is left with a huge amount of spotted information but no critical general view.

Studies that used cell lines were excluded from the literature search. Although this exclusion criterion was aimed at restricting the search to the findings from human tissues, it leads to the exclusion of studies that are crucial to offer a mechanistic interpretation of what observed in the human adrenal tissues. Hence, their findings should be integrated with those reported in the review.

We fully agree with the Reviewer’s comment. However, we decided to exclude both animal and in vitro studies, for two reasons: firstly, as suggested from the flowchart, the amount of evidence which should have been included in this case, would have led to a much more complex manuscript, with a possibly overwhelming amount of information for the reader. Secondly, in in vitro experiments predominantly using immortalized cell lines, additional alterations are present, as possible further confounding factors. In this context, no robust conclusions could be drawn from the combination of the available in vivo and in vitro data. Thus, we preferred omitting this important information about the mechanistic interpretation. We have now included this limitation in the revised version of the manuscript (p. 14, lines 566-573).

Age-related differences are ignored.

We would like to thank the Reviewer for raising this issue. The significance of APCCs in age-related hypertension are now discussed in the manuscript (p. 3-4, lines 126-132).

Similarly, receptors as GPER were not considered at all.

We would like to thank the Reviewer for pointing out this interesting finding. This information is now added in the revised version of the manuscript (p. 8, lines 328-330).

Reviewer 3 Report

In this manuscript, the authors make a large overview of the current knowledge from transcriptomics, epigenetics and metabolomics of primary aldosteronism. As such, a review of this type and breadth is welcome; this notwithstanding, I have several broad comments and series of points raised that need to be addressed in a revised version, as follows:

First, the manuscript as written is heavily and consistently non-idiomatic in terms of English usage; this needs to be addressed by enlisting and acknowledging a native English-speaking colleague to copy-edit a final revised version.

Second, while in some parts the results are discussed and commented on, others are more likely a catalogue. This is for example the case for the modifications of expression of microRNAs or of ion channels. I strongly encourage the authors to flesh out these parts. More particularly, the role of circulating microRNAs as a tool to discriminate unilateral from bilateral form of Primary Aldosteronism has to be discussed (Decman et al, Front Endocrinol, 2019).
Moreover it is now clearly established the ion channels play an important role in the development of primary aldosteronism. In addition to modification in potassium channels expression, the expression of different calcium channels has been investigated in Daniil et al, EBioMed 2016.        

Third, the role of APCC in APA development is still matter of debate. Whereas, different hypotheses were proposed to explain APA formation, The absence of KCNJ5 mutations in APCC could suggest that APA harboring KCNJ5 mutation could have a different origin that the APA harboring other kind of mutations. Recently, De Sousa et al have reported the presence of KCNJ5 mutations in APCC in adrenal with APA. This important result has to be added and discussed.

Fourth, surprisingly the references are not always appropriate and I strongly recommend update.

  • I suggest to reformulate the first sentence of the second paragraph to clarify its general sense: “The most frequent … increasing Ca2+ influx” (p2, lines 54-58)

In the introduction section:

  • It is mentioned that mutations in ATP1A1 are responsible for the increase of the intracellular Ca2+ concentration leading to stimulation of CYP11B2 There is no evidence for an increase of intracellular Ca2+ concentration, it has been suggested that mutations in ATP1A1 lead to H+ leak and cytoplasm acidification that could explain the stimulation of CYP11B2 expression.
  • The ref 10 (p2, line 65) (Tissier et al, Cancer Res, 2005) seems not to be appropriate. In this paper, Tissier et al reported the activation of Wnt/b-catenin signaling pathway due to mutations in b-catenin in both benign and malignant adrenocortical tumors but they do not discuss the role of b-catenin in promoting CYP11B2 synthesis. I rather suggest to use the reference Berthon et al, Hum Mol Genet, 2014 entitled “Wnt/ b-catenin signalling is activated in aldosterone-producing adenomas and controls aldosterone production”.

In the results section:

  • P3 line 105 and line 109, I suggest to add the following ref: Fernandes Rosa et al, Hypertension, 2014 “Genetic spectrum and clinical correlates of somatic mutations in aldosterone-producing adenoma” where extended genotype-phenotype correlations were performed.
  • P3 line 109, I suggest also to add the following references Beuschlein et al, Nat Genet, 2013 “Somatic mutations in ATP1A1 and ATP2B3 lead to aldosterone-producing adenomas and secondary hypertension” and Azizan et al, Nat Genet, 2013 “Somatic mutations in ATP1A1 and CACNA1D underlie a common subtype of adrenal hypertension” where clinical characteristics of patients with ATP1A1 and ATP2B3 were described.
  • P3, line 114, Nishimoto and colleagues were the first to describe APCC in 2010 and Boulkroun and colleagues also in 2010 have described and characterized the foci and megafoci as well as APCC. The references Nishimoto et al, JCEM, 2010 “Adrenocortical zonation in humans under normal and pathological conditions” and Boulkroun et al, Hypertension, 2010 “Adrenal cortex remodeling and functional zona glomerulosa hyperplasia in primary aldosteronism” have to be added.
  • P3, lines 118-125, presence of mutations in KCNJ5 have been recently reported in APCC, this important result has to be added and discussed and the following reference added: De Sousa et al, Hypertension, 2020 “Genetic, cellular and molecular heterogeneity in adrenals with aldosterone-producing adenoma”
  • P12, lines 455-460, measure of hybrid steroids in APA harboring different mutations has also been reported in De Sousa et al, Hypertension, 2020.

There are also some details that have to be change:

  • Replace “neoplasms” by “aldosterone producing adenoma” in the Simple Summary and in the Abstract
  • Throughout the manuscript the gene names have to be in italic.
  • Figure 3 is cited before Figure 2 in the manuscript, this has to be corrected.

Finally, even if it was published after July 2021, it seems important to report and discuss in the G-protein coupled receptors section the recent paper by Zhou et al, Nat Genet, 2021 were the authors reported the presence of somatic mutations in GNAQ/GNA11 in CTNNB1-mutatnt APA presenting in puberty, pregnancy or menopause.

Author Response

Reviewer 3

In this manuscript, the authors make a large overview of the current knowledge from transcriptomics, epigenetics and metabolomics of primary aldosteronism. As such, a review of this type and breadth is welcome; this notwithstanding, I have several broad comments and series of points raised that need to be addressed in a revised version, as follows:

First, the manuscript as written is heavily and consistently non-idiomatic in terms of English usage; this needs to be addressed by enlisting and acknowledging a native English-speaking colleague to copy-edit a final revised version.

We appreciate the Reviewer’s point and in the revised version an attempt was made to engage a colleague as suggested for final proof reading.

Second, while in some parts the results are discussed and commented on, others are more likely a catalogue. This is for example the case for the modifications of expression of microRNAs or of ion channels. I strongly encourage the authors to flesh out these parts. More particularly, the role of circulating microRNAs as a tool to discriminate unilateral from bilateral form of Primary Aldosteronism has to be discussed (Decman et al, Front Endocrinol, 2019).
Moreover it is now clearly established the ion channels play an important role in the development of primary aldosteronism. In addition to modification in potassium channels expression, the expression of different calcium channels has been investigated in Daniil et al, EBioMed 2016.        

We would like to thank the Reviewer for raising these points. As suggested, in the revised version of the manuscript additional information is provided on several topics that were previously briefly discussed and the examples provided were also included and discussed in the text (p. 5, lines 216-220, p. 7, lines 282-285, p. 8, lines 328-330, p. 9, lines 397-412, p. 11-12, lines 431-478, p. 13, lines 518-527 and p. 14, lines 548-553)

Third, the role of APCC in APA development is still matter of debate. Whereas, different hypotheses were proposed to explain APA formation, The absence of KCNJ5 mutations in APCC could suggest that APA harboring KCNJ5 mutation could have a different origin that the APA harboring other kind of mutations. Recently, De Sousa et al have reported the presence of KCNJ5 mutations in APCC in adrenal with APA. This important result has to be added and discussed.

We fully agree with the Reviewer’s comment and added this information in the revised manuscript. The part discussing APCC is now amended, including the new nomenclature (aldosterone producing micronodules-APM). The newly identified presence of KCNJ5 mutations in APCCs, the presence of APCCs in BAH and the debate about the role of APCCs in the development of APAs is now discussed in detail (p. 3-4 , lines 119-179 and p. 14, lines 548-553).

Fourth, surprisingly the references are not always appropriate and I strongly recommend update.

The references have now been crosschecked and where necessary have been updated.

I suggest to reformulate the first sentence of the second paragraph to clarify its general sense: “The most frequent … increasing Ca2+ influx” (p2, lines 54-58)

As suggested by the Reviewer, the sentence is now corrected.

In the introduction section:

It is mentioned that mutations in ATP1A1 are responsible for the increase of the intracellular Ca2+ concentration leading to stimulation of CYP11B2 There is no evidence for an increase of intracellular Ca2+ concentration, it has been suggested that mutations in ATP1A1 lead to H+ leak and cytoplasm acidification that could explain the stimulation of CYP11B2 expression.

We would like to thank the Reviewer for pointing out this inaccuracy, this was corrected in the present version of the manuscript (p. 2, lines 63-66).

The ref 10 (p2, line 65) (Tissier et al, Cancer Res, 2005) seems not to be appropriate. In this paper, Tissier et al reported the activation of Wnt/b-catenin signaling pathway due to mutations in b-catenin in both benign and malignant adrenocortical tumors but they do not discuss the role of b-catenin in promoting CYP11B2 synthesis. I rather suggest to use the reference Berthon et al, Hum Mol Genet, 2014 entitled “Wnt/ b-catenin signalling is activated in aldosterone-producing adenomas and controls aldosterone production”.

As recommended by the Reviewer, the reference was replaced appropriately (now p. 2, line 68).

In the results section:

P3 line 105 and line 109, I suggest to add the following ref: Fernandes Rosa et al, Hypertension, 2014 “Genetic spectrum and clinical correlates of somatic mutations in aldosterone-producing adenoma” where extended genotype-phenotype correlations were performed.

According to the Reviewer’s suggestion, we added the reference in the revised version of the manuscript (now p. 3, line 112 and 117).

P3 line 109, I suggest also to add the following references Beuschlein et al, Nat Genet, 2013 “Somatic mutations in ATP1A1 and ATP2B3 lead to aldosterone-producing adenomas and secondary hypertension” and Azizan et al, Nat Genet, 2013 “Somatic mutations in ATP1A1 and CACNA1D underlie a common subtype of adrenal hypertension” where clinical characteristics of patients with ATP1A1 and ATP2B3 were described.

We would like to thank the Reviewer for his suggestion. The respective references are added in the present version of the manuscript (now p. 3, line 117).

P3, line 114, Nishimoto and colleagues were the first to describe APCC in 2010 and Boulkroun and colleagues also in 2010 have described and characterized the foci and megafoci as well as APCC. The references Nishimoto et al, JCEM, 2010 “Adrenocortical zonation in humans under normal and pathological conditions” and Boulkroun et al, Hypertension, 2010 “Adrenal cortex remodeling and functional zona glomerulosa hyperplasia in primary aldosteronism” have to be added.

Following the Reviewer’s suggestion, the appropriate references are added in the text (p. 3, line 126).

P3, lines 118-125, presence of mutations in KCNJ5 have been recently reported in APCC, this important result has to be added and discussed and the following reference added: De Sousa et al, Hypertension, 2020 “Genetic, cellular and molecular heterogeneity in adrenals with aldosterone-producing adenoma”

We would like to thank the Reviewer for pointing out this important finding. This is now discussed in the present version of the manuscript, and the respective reference is added (now p. 4, lines 143-146).

P12, lines 455-460, measure of hybrid steroids in APA harboring different mutations has also been reported in De Sousa et al, Hypertension, 2020.

This interesting finding is now also added in the respective section of the manuscript (now p. 13, lines 518-522).

There are also some details that have to be change:

Replace “neoplasms” by “aldosterone producing adenoma” in the Simple Summary and in the Abstract

This term has been replaced, as suggested.

Throughout the manuscript the gene names have to be in italic.

Gene abbreviations are now in italic throughout the text.

Figure 3 is cited before Figure 2 in the manuscript, this has to be corrected.

The Figures are now in the correct order.

Finally, even if it was published after July 2021, it seems important to report and discuss in the G-protein coupled receptors section the recent paper by Zhou et al, Nat Genet, 2021 were the authors reported the presence of somatic mutations in GNAQ/GNA11 in CTNNB1-mutatnt APA presenting in puberty, pregnancy or menopause.

We would like to thank the Reviewer for this remark. This important finding is now added in the revised version of the manuscript (p. 4, lines 146-151).

Round 2

Reviewer 2 Report

In the revised version of the manuscript information are reported in more critical way. However, more effort could be done in guiding the reader in dissecting literature and catching the noveltis.